# Sex-specific differences in children attending the emergency department: prospective observational study

Joany M Zachariasse,[1] Dorine M Borensztajn,[1] Daan Nieboer,[2] Claudio F Alves,[3] Susanne Greber-Platzer,[4] Claudia M G Keyzer-Dekker,[5] Ian K Maconochie,[6] Ewout W Steyerberg,[7] Frank J Smit,[8] Henriëtte A Moll [1]

For numbered affiliations see end of article.

**Correspondence to**
Dr Henriëtte A Moll;
h.a.moll@erasmusmc.nl

## ABSTRACT

**Objective** To assess the role of sex in the presentation and management of children attending the emergency department (ED).

**Design** The TrIAGE project (TRiage Improvements Across General Emergency departments), a prospective observational study based on curated electronic health record data.

**Setting** Five diverse European hospitals in four countries (Austria, The Netherlands, Portugal, UK).

**Participants** All consecutive paediatric ED visits of children under the age of 16 during the study period (8–36 months between 2012 and 2015).

**Main outcome measures** The association between sex (male of female) and diagnostic tests and disease management in general paediatric ED visits and in subgroups presenting with trauma or musculoskeletal, gastrointestinal and respiratory problems and fever. Results from the different hospitals were pooled in a random effects meta-analysis.

**Results** 116 172 ED visits were included of which 63 042 (54%) by boys and 53 715 (46%) by girls. Boys accounted for the majority of ED visits in childhood, and girls in adolescence. After adjusting for age, triage urgency and clinical presentation, girls had more laboratory tests compared with boys (pooled OR 1.10, 95% CI 1.05 to 1.15). Additionally, girls had more laboratory tests in ED visits for respiratory problems (pooled OR 1.15, 95% CI 1.04 to 1.26) and more imaging in visits for trauma or musculoskeletal problems (pooled OR 1.10, 95% CI 1.01 to 1.20) and respiratory conditions (pooled OR 1.14, 95% CI 1.05 to 1.24). Girls with respiratory problems were less often treated with inhalation medication (pooled OR 0.76, 95% CI 0.70 to 0.83). There was no difference in hospital admission between the sexes (pooled OR 0.99, 95% CI 0.95 to 1.04).

**Conclusion** In childhood, boys represent the majority of ED visits and they receive more inhalation medication. Unexpectedly, girls receive more diagnostic tests compared with boys. Further research is needed to investigate whether this is due to pathophysiological differences and differences in disease course, whether girls present signs and symptoms differently, or whether sociocultural factors are responsible.

## INTRODUCTION

There is increasing evidence that differences between men and women are important in

## Strengths and limitations of this study

► The present study was based on a large, multicentre and international cohort of more than 100 000 emergency department (ED) visits.
► The study is the first to assess the role of sex in the prevalence and type of clinical presentations and in the diagnostics and management of children attending the ED.
► Due to the use of standardised, routinely collected data, only general patterns and broad subgroups of patients could be evaluated.
► Our cohort of ED visits included repeat visits from the same child, which could have led to underestimation of the SEs and potentially to wider CIs.
► Only two hospitals in our study included patients with major trauma, which limits generalisability to this subgroup of patients.

the epidemiology, pathophysiology, treatment and outcome of many diseases.[1–3] The Institute of Medicine has emphasised that sex is an important variable that should be considered in biomedical and health related research.[4] Sex refers to the biological differences between men and women, while gender refers to a broader concept including social and cultural distinctions associated with a given sex.[3] In the area of emergency medicine, sex and gender-specific differences have been found in adults, in topics as broad as the clinical presentation and outcomes of acute myocardial infarction,[5] the prevalence and survival of out-of-hospital cardiac arrest[6] and the epidemiology of sports-related injuries.[7]

In children, research on the impact of sex and gender on health is scarce, particularly in the field of emergency medicine. Only 22 relevant articles described sex or gender differences in children related to emergency medicine between January 2000 and November 2017, according to PubMed (online supplemental material: appendix 1). Of these, only two studies described

differences between sexes in overall emergency department (ED) resource use or management. Both studies observed higher ED attendance rates in boys compared with girls, but did not address specific types of clinical presentations or evaluate differences in disease management or outcome.[8 9] Other studies focused on specific disease groups such as trauma and injuries, mental health related conditions and asthma. In these areas studies report conflicting results regarding differences in the rate of ED visits between boys and girls. Diagnosis and management were poorly studied. Only one study in children with asthma looked at differences in ED management and did not find a difference in treatment between boys and girls.[10]

The aim of the current study is to assess the role of sex in the clinical presentation, diagnostics and management in the general population of children attending the ED. Through this study, we aim to gain more insight in sex-specific differences in paediatric emergency medicine and identify areas for future research.

## METHODS

This study was embedded in the TrIAGE project (TRiage Improvements Across General Emergency departments), a prospective observational study, based on electronic health record data, in five EDs in four European countries (The Netherlands, UK, Austria, Portugal).

### Study settings and patient population

In the TrIAGE project, 119 209 consecutive ED visits of children and adolescents under the age of 16 years were included. Enrolment took place during a period of 8–36 months between 2012 and 2015 in five diverse study sites (online supplemental material: appendix 2). We restricted our analysis to ED visits with complete triage data, thereby excluding all visits with missing triage categories or missing presenting problem. The General Hospital Vienna only included ED visits for medical disorders because the majority of trauma patients were seen in the department of traumatology. A small proportion of low urgent trauma cases were still seen in the ED by the 'in-house paediatrician' and we excluded these remaining patients to reduce selection bias.

### Data collection

The TrIAGE project is based on routinely collected, standardised electronic health record data. At the start of the study, a set of minimally required variables was determined. Completeness and accuracy of the data was discussed during a site visit and telephone meetings using a checklist quality control. With posters, newsletters and presentations, nurses were encouraged to complete the full medical records including vital sign measurements.

During the study period, nurses and physicians entered the clinical data in the medical or nursing records. After data extraction, careful data harmonisation and quality checks were performed.

### Determinants

In all participating hospitals, sex (male or female) and age were routinely registered for each patient as part of the administrative process.

We used triage data to determine type of presenting problem and triage urgency. All participating hospitals used the Manchester Triage System (MTS). This flowchart-based emergency medical triage system is the most commonly used triage system in Europe. In the MTS, the triage nurse is required to select a flowchart for each patient, representing the primary symptom, such as *Shortness of Breath* or *Wounds*. Each flowchart consists of signs and symptoms named discriminators that are ranked by priority. The nurse then gathers information on the discriminators from top to bottom. Selection of a discriminator allocates the patient to the related urgency category ranging from 'immediate' (0 min maximum waiting time) to 'non-urgent' (240 min maximum waiting time).[11]

Age, triage urgency and clinical presentation were considered as potential confounding variables in the relationship between sex and disease management. Age was maintained

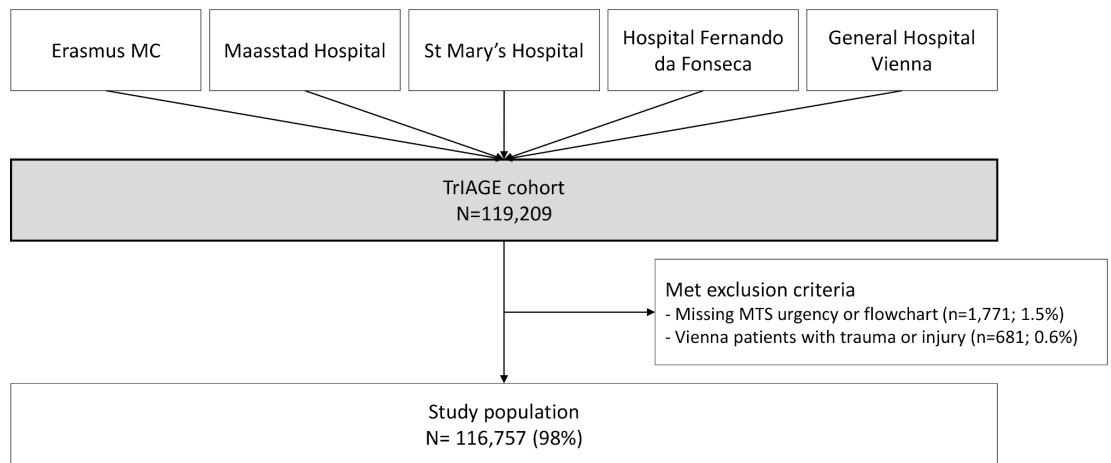

**Figure 1** Flow diagram of the study population. MTS, Manchester Triage System; TrIAGE, TRiage Improvements Across General Emergency departments.

**Table 1** Baseline characteristics of the study population

| | Boys, n=63 042 | Girls, n=53 715 |
|---|---|---|
| **Age, no. (%)** | | |
| 0–<1 years | 10 289 (16) | 8009 (15) |
| 1–<12 years | 44 233 (70) | 36 929 (69) |
| ≥12 years | 8520 (14) | 8777 (16) |
| **MTS urgency, no. (%)** | | |
| Emergent/very urgent | 7532 (12) | 5232 (10) |
| Urgent | 17 583 (28) | 14 839 (28) |
| Standard/non-urgent | 37 927 (60) | 33 644 (63) |
| **Clinical presentation, no (%)** | | |
| Cardiac | 688 (1) | 710 (1) |
| Dermatological | 8179 (13) | 6443 (12) |
| Ear, nose and throat | 5808 (9) | 5754 (11) |
| Gastrointestinal | 9346 (15) | 8871 (17) |
| Neurological or psychiatric | 2437 (4) | 2254 (4) |
| Respiratory | 8218 (13) | 5742 (11) |
| Trauma or musculoskeletal | 11 587 (18) | 9045 (17) |
| General malaise | 12 934 (21) | 11 412 (21) |
| Urological or gynaecological | 1315 (2) | 1268 (2) |
| Other | 2530 (4) | 2216 (4) |
| **Diagnostics, no. (%)** | | |
| Laboratory | 12 461 (20) | 11 597 (22) |
| Imaging | 13 386 (21) | 11 216 (21) |
| **Medication, no. (%)** | | |
| Inhalation | 5177 (8) | 3389 (6) |
| Intravenous | 4901 (8) | 4143 (8) |
| **Disposition, n (%)** | | |
| Mortality at the ED | 12 (<0.1) | 4 (<0.1) |
| ICU admission | 369 (0.6) | 299 (0.6) |
| Hospital admission | 6447 (10) | 4965 (9) |
| Discharge/other | 56 214 (89) | 48 447 (90) |

Percentages are column totals.
ED, emergency department; ICU, intensive care unit; MTS, Manchester Triage System.

as a continuous variable. Triage urgency was defined as the MTS urgency category and placed into three groups: high urgent (MTS category immediate or high urgent), urgent (MTS category urgent) and low urgent (MTS category standard or non-urgent). Three hospitals (Erasmus MC, General Hospital Vienna and Hospital Fernando da Fonseca) implemented MTS modifications. To ensure consistency among hospitals, the urgency levels according to MTS version 3 with Dutch modifications for children with fever were modelled in all hospitals.[11][12] We distinguished ten clinical presentations, based on the main problem presented during triage according to MTS flowchart (online supplemental material: appendix 3).

## Data analysis
In descriptive analyses, differences between male and female sex in the type and severity of problems with which they presented at the ED were explored.

Multivariable logistic regression models were used to assess the association between sex and diagnostics and management in the ED for each study site. We selected laboratory tests and imaging as important markers for diagnostics, and inhalation medication, intravenous medication or fluids and hospital admission as markers for disease management. Analyses were adjusted for age, triage urgency and clinical presentation, and boys were determined as the reference group. ED visits were treated as independent in the analyses although some children likely had repeated visits and those visits could not be distinguished in this study. The results of the individual hospitals were pooled in a random effects meta-analysis and presented in forest plots. ORs were considered statistically significant when the 95% CI did not include 1. Heterogeneity among settings was assessed using the Cochran's Q statistic and the $I^2$ statistic.

We performed subgroup analyses in subgroups of children presenting with trauma or musculoskeletal problems, children presenting with fever, children presenting with gastro-intestinal problems and children presenting with shortness of breath. Fever was defined as either a temperature of ≥38.5°C at the ED or the item 'fever' selected as triage discriminator. Shortness of breath, gastrointestinal and trauma or musculoskeletal problems were defined based on the triage flowchart. We selected these four clinical presentations because they were among the largest subgroups of patients and represent important clinical entities in children.

Statistical analyses were performed in SPSS Statistics V.21.0 (IBM Corp, Armonk, NY, USA) and R V.3.4.2 (R Foundation, Vienna, Austria). The package metafor was used to conduct the random effects meta-analysis and create forest plots.

### Patient and public involvement
There was no patient involvement in the design of the study or the writing of the manuscript. We plan to disseminate the results to patient organisations, clinicians and policy makers working in emergency care on publication.

## RESULTS
Of all 119 209 ED visits included in the TrIAGE cohort, 116 757 ED visits (98%) were included in the study (figure 1). There was no significant difference between the sexes in the proportion of excluded ED visits (Pearson's $\chi^2(1)$=0.08, p value 0.77). General characteristics of the study population are presented in table 1.

### Relation between sex and age, triage urgency and clinical presentation
In the total study population, the sex imbalance in number of ED visits changed with age. In childhood, consistently more boys than girls attended the ED, and the

**Table 2** Sex-specific differences in age, triage and clinical presentation

| | Boys, n=63042 | Range over hospitals (%) | Girls, n=53715 | Range over hospitals (%) |
|---|---|---|---|---|
| **Age** | | | | |
| 0–<1 y | 10289 (56) | 53–60 | 8009 (44) | 40–47 |
| 1–<12 y | 44233 (54) | 53–59 | 36929 (46) | 41–47 |
| ≥12 y | 8520 (49) | 43–54 | 8777 (51) | 46–57 |
| **Triage urgency** | | | | |
| Emergent/very urgent | 7532 (59) | 54–63 | 5232 (47) | 37–46 |
| Urgent | 17583 (54) | 51–58 | 14839 (46) | 42–49 |
| Standard/non-urgent | 37927 (53) | | 33644 (41) | |
| **Clinical presentation** | | | | |
| Cardiac | 688 (49) | 46–61 | 710 (51) | 39–54 |
| Dermatological | 8179 (56) | 53–61 | 6443 (44) | 39–47 |
| Ear, nose and throat | 5808 (50) | 48–59 | 5754 (50) | 41–52 |
| Gastrointestinal | 9346 (51) | 49–57 | 8871 (49) | 43–51 |
| Neurological or psychiatric | 2437 (52) | 50–55 | 2254 (48) | 45–50 |
| Respiratory | 8218 (59) | 57–64 | 5742 (41) | 36–43 |
| Trauma or musculoskeletal | 11587 (56) | 54–59 | 9045 (44) | 41–46 |
| General malaise | 12934 (53) | 52–56 | 11412 (47) | 44–48 |
| Urological or gynaecological | 1315 (51) | 40–80 | 1268 (49) | 20–60 |
| Other | 2530 (53) | 52–56 | 2216 (47) | 44–49 |

Percentages are row totals.

ratio reversed in adolescence (table 2). Boys accounted for the majority of visits in all triage urgency levels, most evidently in the high urgency categories. They had the highest proportion of ED visits with dermatological, respiratory and traumatic symptoms, while in children presenting with cardiac, ear, nose and throat, gastrointestinal and neurological conditions, the boy to girl ratio was almost equal.

We further explored the relation between sex and triage urgency. The higher rate of boys' ED visits in childhood and girls' ED visits in adolescence was consistent among the different urgency categories. Also, the sex-specific differences in the different types of clinical presentation were similar in the high, intermediate and low urgency categories. An exception was the subgroup of urological and gynaecological presentations where there were almost only boys prioritised as high urgency. Also notable was the subgroup of respiratory conditions that represents the largest population of high urgency visits with a 1.5 times higher proportion of boys compared with girls (online supplemental material: appendix 4).

**Sex-specific differences in diagnostics at the ED**
In the general population of children visiting the ED, adjusted for age, triage urgency and clinical presentation, girls had more laboratory tests at the ED than boys (pooled OR 1.10, 95% CI 1.05 to 1.15) (table 3, figure 2). This finding was consistent in all hospitals, although there was some evidence for heterogeneity (Q-statistic

7.27, p=0.12, $I^2$ statistic 45.8%). Significant higher rates of laboratory testing in girls were also seen in the subgroup of ED visits for respiratory problems (pooled OR 1.15, 95% CI 1.04 to 1.26) and trends were observed in ED visits for trauma or musculoskeletal problems (pooled OR 1.12, 95% CI 0.94 to 1.35) and fever (pooled OR 1.12, 95% CI 0.98 to 1.27) (online supplemental material: appendix 5).

In the overall population, no differences in imaging between the sexes were found. However, in the subgroups of ED visits for trauma or musculoskeletal problems and respiratory conditions, more imaging was conducted in girls (pooled OR 1.10, 95% CI 1.01 to 1.20 and pooled OR 1.14, 95% CI 1.05 to 1.24, respectively).

**Sex-specific differences in management at the ED**
In the general population of children attending the ED, adjusted for age, triage urgency and clinical presentation, there was no difference in the rate of IV medication or fluids (OR 0.96, 95% CI 0.89 to 1.03) and in the rate of hospital admission (OR 0.99, 95% CI 0.95 to 1.04) (table 3, figure 2). Neither was this difference seen in one of the subgroups. Remarkably, there was a large difference in the use of inhalation medication in the subgroup of ED visits for respiratory conditions. In this subgroup, girls received less inhalation medication (pooled OR 0.76, 95% CI 0.70 to 0.83) (online supplemental material: appendix 5).

**Table 3** Associations of sex with emergency department (ED) diagnostics and management in the total paediatric ED population and in subgroups based on type of clinical presentation (boys as reference group)

| | | | Heterogeneity | | |
| --- | --- | --- | --- | --- | --- |
| | | **Pooled OR (95% CI)*** | **I² (%)** | **Q** | **P value** |
| **Total paediatric ED population** | | | | | |
| n=116 757 | | | | | |
| **Laboratory tests** | **Girls** | **1.10 (1.05 to 1.15)** | **45.8** | **7.27** | **0.12** |
| | **Boys** | **Reference** | | | |
| Imaging | Girls | 1.01 (0.98 to 1.04) | 0 | 1.84 | 0.77 |
| | Boys | Reference | | | |
| Intravenous medication or fluids | Girls | 0.96 (0.89 to 1.03) | 50.5 | 8.26 | 0.08 |
| | Boys | Reference | | | |
| Hospital admission | Girls | 0.99 (0.95 to 1.04) | 0 | 3.38 | 0.50 |
| | Boys | Reference | | | |

| | | | Heterogeneity | | |
| --- | --- | --- | --- | --- | --- |
| | | **Pooled OR (95% CI)†** | **I² (%)** | **Q** | **P value** |
| **Trauma or musculoskeletal** | | | | | |
| n=20 632 | | | | | |
| Laboratory tests | Girls | 1.12 (0.94 to 1.35) | 39.8 | 4.76 | 0.19 |
| | Boys | Reference | | | |
| **Imaging** | **Girls** | **1.10 (1.01 to 1.20)** | **44.9** | **5.35** | **0.15** |
| | **Boys** | **Reference** | | | |
| Intravenous medication or fluids | Girls | 0.84 (0.62 to 1.12) | 74.2 | 12.64 | 0.01 |
| | Boys | Reference | | | |
| Hospital admission | Girls | 0.90 (0.77 to 1.05) | 37.7 | 4.76 | 0.19 |
| | Boys | Reference | | | |
| **Gastrointestinal** | | | | | |
| n=18 217 | | | | | |
| Laboratory tests | Girls | 1.05 (0.90 to 1.21) | 71.6 | 12.74 | 0.01 |
| | Boys | Reference | | | |
| Imaging | Girls | 0.95 (0.85 to 1.05) | 19.1 | 5.23 | 0.26 |
| | Boys | Reference | | | |
| Intravenous medication or fluids | Girls | 0.92 (0.79 to 1.07) | 53.6 | 8.44 | 0.08 |
| | Boys | Reference | | | |
| Hospital admission | Girls | 0.96 (0.87 to 1.05) | 0 | 1.66 | 0.80 |
| | Boys | Reference | | | |
| **Respiratory** | | | | | |
| n=13 960 | | | | | |
| **Laboratory tests** | **Girls** | **1.15 (1.04 to 1.26)** | **0** | **2.51** | **0.64** |
| | **Boys** | **Reference** | | | |
| **Imaging** | **Girls** | **1.14 (1.05 to 1.24)** | **0** | **2.69** | **0.61** |
| | **Boys** | **Reference** | | | |
| **Inhalation medication** | **Girls** | **0.79 (0.73 to 0.86)** | **0** | **2.38** | **0.67** |
| | **Boys** | **Reference** | | | |
| Intravenous medication or fluids | Girls | 1.10 (0.96 to 1.26) | 0 | 2.08 | 0.72 |
| | Boys | Reference | | | |

Continued

**Table 3** Continued

| | | | Heterogeneity | | |
| | | Pooled OR (95% CI)† | I$^2$ (%) | Q | P value |
| --- | --- | --- | --- | --- | --- |
| Hospital admission | Girls | 1.10 (1.00 to 1.22) | 0 | 2.32 | 0.72 |
| | Boys | Reference | | | |
| Fever | | | | | |
| n=8782 | | | | | |
| Laboratory tests | Girls | 1.12 (0.98 to 1.27) | 42.8 | 6.88 | 0.14 |
| | Boys | Reference | | | |
| Imaging | Girls | 1.03 (0.91 to 1.16) | 0 | 1.81 | 0.77 |
| | Boys | Reference | | | |
| Intravenous medication or fluids | Girls | 1.04 (0.92 to 1.19) | 0 | 3.15 | 0.53 |
| | Boys | Reference | | | |
| Hospital admission | Girls | 1.07 (0.96 to 1.20) | 0 | 2.61 | 0.63 |
| | Boys | Reference | | | |

Bold represents Odds Ratios that were considered statistically significant when the 95% confidence interval did not include 1.

*Summary OR based on a random effects model where results from individual settings are pooled. Model adjusted for age, triage urgency and clinical presentation.

†Summary OR based on a random effects model where results from individual settings are pooled. Model adjusted for age and triage urgency.

## DISCUSSION

In a European cohort of more than 100 000 ED visits, boys accounted for the majority of ED visits in childhood, while in adolescence, the male to female ratio reversed. We found evidence that more diagnostic tests were conducted in girls compared with boys: girls had more laboratory tests overall (pooled OR 1.10, 95% CI 1.05 to 1.15), and in the subgroup of ED visits for respiratory problems (pooled OR 1.15, 95% CI 1.04 to 1.26) while girls received more imaging in the ED visits for trauma or musculoskeletal problems (pooled OR 1.10, 95% CI 1.01 to 1.20) and for respiratory conditions (pooled OR 1.14, 95% CI 1.05 to 1.24). Girls, however, received less inhalation medication when presenting with respiratory problems (pooled OR 0.76, 95% CI 0.70 to 0.83). There was no difference between sexes in the odds of hospital admission (OR 0.99, 95% CI 0.95 to 1.04).

This is the first study that assesses the role of sex in paediatric emergency medicine, using a large, multicentre and international cohort. Despite the diversity in study sites, results were comparable across the participating EDs, supporting generalisability of the findings.

The higher attendance rate of boys in European EDs is consistent with findings from the US National Hospital Ambulatory Medical Surveys.[13] Diverse biological explanations have been proposed, such as an increased susceptibility to infections and higher risk of lung disease in men, due to male sex hormones.[14–16] Behavioural, social and cultural factors might lead to differences in the number and types of ED visits as well. These may include differences in activities, risk taking behaviour, parenting practices, and different referral decisions by primary care

providers.[17 18] Sex hormones have been considered responsible for differences in lung development and differences in prevalence and severity of lung diseases.[14 15 19] A higher prevalence of asthma in boys during childhood has previously been reported, although reports in emergency care have been inconsistent.[16 20] However, boys' increased susceptibility to respiratory conditions could indicate a more severe disease course, possibly explaining the higher rate of inhalation medications used in our study.

Beside sex differences in disease presentations, our study found differences in diagnostics between boys and girls. We did not find any previous publication with a similar finding. Sex-specific differences in the need for diagnostic tests could be caused by differences in disease severity, but may also be attributed to differences in clinical presentation. Boys and girls may differ regarding the presenting signs and symptoms, the way they express pain or distress, or their ability to phrase their symptoms. Diagnostic uncertainty due to a specific disease presentation may warrant more diagnostic tests. Finally, it is possible that provider attitude subconsciously differs in boys compared with girls. In adults, implicit gender bias in diagnostics and treatment has been found in observational research and standardised case scenarios.[21 22] Our analyses are exploratory and our findings require further research. Future studies are needed that address sex-specific differences in the conducting of diagnostic tests. These studies should aim to elucidate whether differences in the need for diagnostics are due to pathophysiological mechanisms and differences in disease course, whether girls present signs and symptoms differently compared with

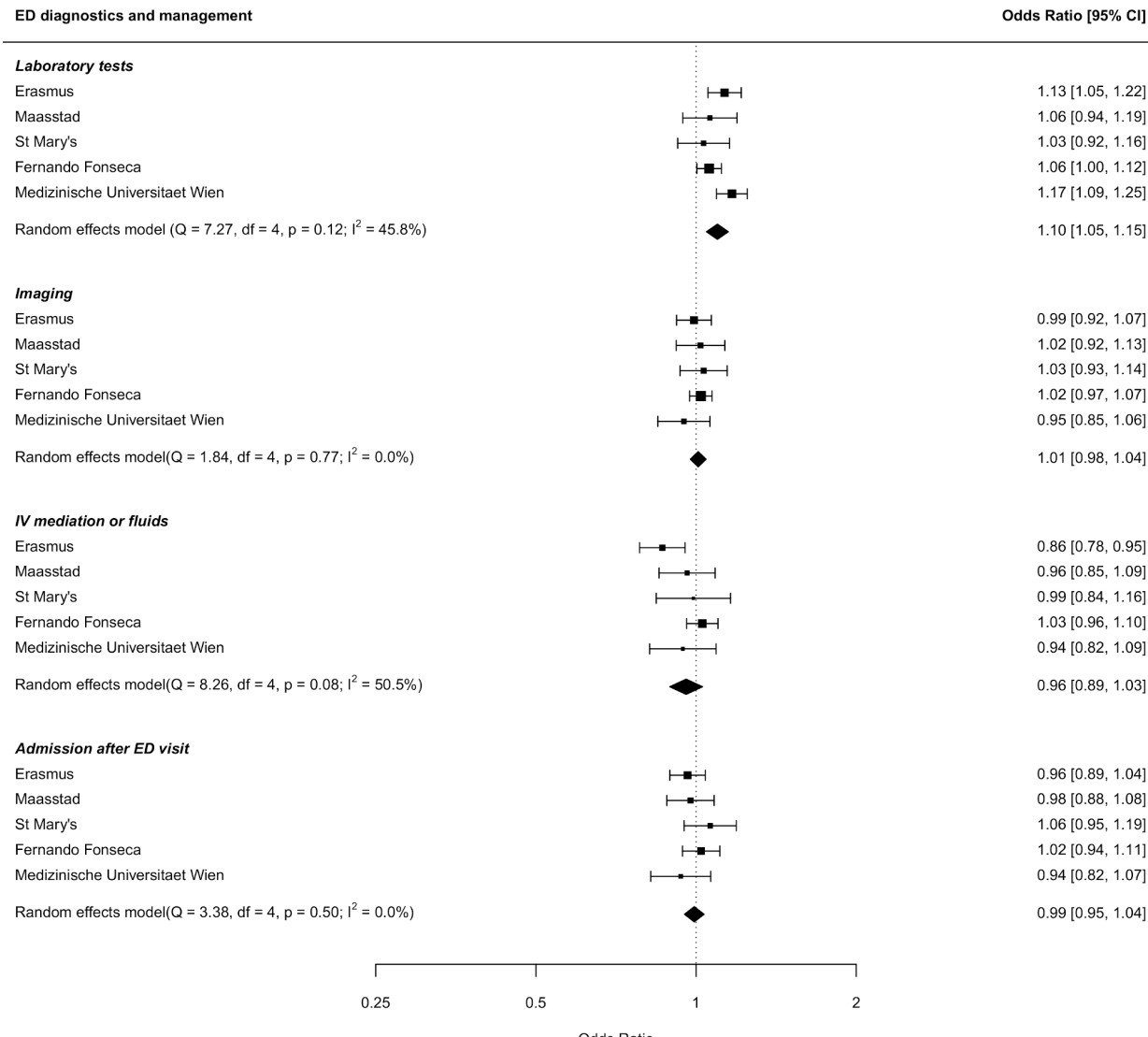

**Figure 2** Associations of sex with ED diagnostics and management in the total study population, adjusted for age, triage urgency and clinical presentation (boys as reference group). ED, emergency department.

boys, or whether there are other sociocultural factors responsible.

## Limitations

These results should be interpreted in light of the limitations of using standardised routinely collected data. Detailed information such as specific types of diagnoses or results from diagnostic tests was not available. Therefore, only general patterns and broad subgroups of patients were evaluated. Furthermore, it is possible that observed differences in diagnostics and management between boys and girls are confounded by sex-specific differences in type and severity of presenting conditions. The analyses were adjusted for age, triage urgency and clinical presentation, and the findings were analysed in specific subtypes of clinical presentations. There may, however, be remaining variation not captured in urgency and presenting problem. Additionally, our cohort of ED visits included repeat visits from the same child which are not independent. In general, ignoring correlated data does not influence effect estimates such as ORs, but it can lead bias in the SEs.[23] By not taking into account the correlation of visits from the same subject we are likely to underestimate the SEs of our effect sizes. Such underestimation could potentially lead to smaller confidence intervals than would be obtained when analyses are adjusted for repeat visits from the same child. A previous study shows a revisit rate of approximately 21% in a representative sample of data from US EDs.[24] Finally, only two hospitals in our study included patients with major trauma which limits generalisability to this subgroup of patients.

## CONCLUSION

In children, the role of sex and gender on health is largely unknown and research assessing sex -specific differences is scare. Our study found that in childhood boys more often present to the ED compared with girls, while in

adolescence this ratio is reversed. The higher need for inhalation medication in boys may represent a higher susceptibility for or a more severe course of respiratory infections. Unexpectedly, girls receive more diagnostic tests compared with boys. Future studies should focus on the role of sex and gender in specific conditions and determine whether there are pathophysiological differences in disease course and severity, whether girls present signs and symptoms differently or whether there are social and cultural factors responsible.

**Author affiliations**
[1] Department of General Paediatrics, Erasmus MC- Sophia Children's Hospital, Rotterdam, The Netherlands
[2] Department of Public Health, Erasmus MC - University Medical Center, Rotterdam, The Netherlands
[3] Department of Paediatrics, Emergency Unit, Hospital Professor Doutor Fernando da Fonseca, Amadora, Portugal
[4] Department of Pediatrics and Adolescent Medicine, Medical University, Vienna, Austria
[5] Department of Pediatric Surgery, Erasmus MC- Sophia Children's Hospital, Rotterdam, The Netherlands
[6] Department of Pediatric Emergency Medicine, Imperial College NHS Healthcare Trust, London, UK
[7] Department of Medical Statistics and Bioinformatics, Leiden University Medical Center, Leiden, The Netherlands
[8] Department of Paediatrics, Maasstad Hospital, Rotterdam, The Netherlands

**Acknowledgements** We gratefully acknowledge Rikke Jorgensen, paediatric research nurse (St. Mary's Hospital, London) and Pinky Rose Espina, paediatric resident (General Hospital, Vienna) for their assistance with data collection for this study.

**Contributors** JMZ contributed to the conceptualisation and design of the study, carried out the analyses, and drafted the initial manuscript. DMB, FJS, CFA, CMGK-D, IKM and SG-P contributed to the conceptualisation and design of the study, coordinated the data collection, contributed to interpretation of the data and critically reviewed the manuscript. DN and EWS contributed to the conceptualisation and design of the study, supervised the analyses and critically reviewed the manuscript. HAM contributed to the conceptualisation and design of the study, contributed to interpretation of the data, critically reviewed the manuscript and supervised the study. All authors approved the final manuscript as submitted and agree to be accountable for all aspects of the work.

**Funding** The authors have not declared a specific grant for this research from any funding agency in the public, commercial or not-for-profit sectors.

**Competing interests** None declared.

**Patient consent for publication** Not required.

**Ethics approval** The study was approved by the medical ethical committees of the participating institutions: Medical Ethics Committee Erasmus MC (MEC-2013-567), Board of Directors Maasstad Ziekenhuis (L2013-103), Imperial College London Joint Research Compliance Office (14/WA/1051), Comissão de Ética para a Saúde do Hospital Prof. Dr. Fernando Fonseca EPE (Estudo Clínico TrIAGE – Parecer Favorável), Ethik Kommission Medizinische der Medizinischen Unversität Wien (EK Nr: 1405/2014). All waived the requirement for informed consent.

**Provenance and peer review** Not commissioned; externally peer reviewed.

**Data availability statement** Data are available upon reasonable request. Data from this study are available upon request to the corresponding author of the study (h.a.moll@erasmusmc.nl), subject to local rules and regulations.

**ORCID iD**
Henriëtte A Moll http://orcid.org/0000-0001-9304-3322

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
