## [Reviewer comments · BMJ Open]

ARTICLE DETAILS

TITLE (PROVISIONAL)	Sex-Specific Differences in Children Attending the Emergency Department: Prospective Observational Study
AUTHORS	Zachariasse, Joany; Borensztajn, Dorine; Nieboer, Daan; Alves, Claudio; Greber-Platzer, Susanne; Keyzer-Dekker, Claudia; Maconochie, Ian; Steyerberg, Ewout; Smit, Frank; Moll, Henriette

VERSION 1 – REVIEW

REVIEWER	Rhonda J. Rosychuk University of Alberta, Canada
REVIEW RETURNED	12-Dec-2019

GENERAL COMMENTS	Sex-Specific Differences in Children Attending the Emergency Department: Prospective Observational Study Zachariasse et al This study is interesting and important. It focuses on sex differences for children presenting at EDs, where there is a paucity of literature. The authors use data from multiple countries, which is a strength, and pool results through random effects meta-analysis. The large data set and analyses by general medical problems and trauma and injuries are also strengths. The manuscript is well written and the figures, tables, and appendices are appropriate. Specific comments: 1. There is a lack of clarity about the unit of the analysis. The authors sometimes reference children and sometimes reference ED visits. Is the unit of analysis the child or the ED visit? It would seem that the unit of analysis is the child, but I would expect that some of these children have more than one ED visit. The Methods section describes 119,209 consecutive ED visits and 1,771 children with incomplete data are removed. The Results section mentions 116,440 children: $116440 + 1771 = 118211 \neq 119209$ so there must be some children who have repeat ED visits. The ED visits from the same child would be correlated, and not be independent data. Hence the standard errors are likely a little too small because correlated data is unaccounted for. Given the large scale of the dataset, the resulting ORs and 95% CIs would not be altered very much. Perhaps authors can restrict to selecting one ED visit per patient so that they do not have to use statistical methods for correlated data.2. Laboratory tests, imaging, medications, and admissions would be related to the presenting condition and not all of these diagnostic/management variables would be relevant
---

	for particular presenting conditions. It would seem to me that models for these outcomes should be adjusted by presenting condition as well. Would not the authors want to know if differences in diagnostics/management occur when adjusted for age, triage, and presenting condition? The subgroup analysis address this a bit but not completely.  3. Analyses were adjusted for age and triage urgency. Was triage urgency considered as an outcome as well to see if there are sex differences in the assigning of triage urgency? This additional analysis could still be adjusted by age. It is unclear how subjective the triage urgency could be. 4. Would be helpful to put the study period in the abstract or at least some indication of the general time period of data because the data from different sites may have been collected at different time periods. 5. I would have expected a bit more summary and/or comparison with the findings of other relevant studies rather than providing an appendix of search terms/citations. 6. It is not clear how the Manchester Triage System urgency category is defined. A reference is not provided and the flowchart in Appendix 3 does not indicate urgency. 7. The grouping of the disposition categories are unclear and/or unjustified. ICU and mortality at ED are combined and I would think that these two outcomes are different. Did the authors mean that all ICU patients died in the ED? I do not see a CONSORT checklist submitted in the documents I can see online.
--	---

REVIEWER	Taneisha Wilson Brown University, USA
REVIEW RETURNED	14-Jan-2020

GENERAL COMMENTS	Page 1: Line 21/27 we use girls and boys here interchangeably with male/female for line 21 would suggest maybe use boy/girl as parenthetical form Page 9: I would suggest a flow diagram here. Line 51, again would use boy vs. girl instead of male-female based on your demographic population. Page 12: line 29-30, thought boys were the reference variable and girls the comparator. If this is the case I would reframe this (Appendix 6). Also, assuming that the age subgroups include >12=adolescent based on page 12 line 53, but this is not explicitly stated.
--

REVIEWER	Tadahiro Goto The University of Tokyo, Japan
REVIEW RETURNED	28-Jan-2020

GENERAL COMMENTS	I appreciate the opportunity to review this manuscript. In this large, multicenter and international cohort study of 116,440 emergency department visits made by children reports that sex is associated with differences in disease presentation and management. While
---

	this study might provide some information for clinicians and researchers, several concerns should be addressed. General comments ===== 1. While this paper is well-written and the data might be informative, I am not sure the importance of the study findings and its implications. What is the knowledge gap? What will be changed by the study findings? The descriptive findings are interesting and should be the important basis for the future investigation. However, the importance and clinical implications of the associations (e.g., boys received more inhalation medication) are not clear. Specific comments INTRODUCTION: 2. Please clarify the knowledge gap. While there are 22-relevant articles in this area, the knowledge gap remains unclear. METHODS SECTION: 3. While authors excluded patients with missingness, the lack of clinical information might be not at random (e.g., critically-ill children). Are there any potential bias due to the missingness? 4. Why authors used laboratory tests and imaging as an outcome? 5. Why authors choose fever and shortness of breath for the subgroup analysis? I understand that these two complaints are common, but it looks arbitral when there are no supporting literature or rational. 6. Authors performed subgroup analysis based on patient's chief complaints. This implies that the association between sex and outcomes may differ across chief complaints. If so, why authors did not include chief complains as an adjustment variable? RESULTS SECTION: 7. Page 12, line 29-31. "Boys, on the other hand, received more inhalation medication (pooled OR 0.75...)." The subject and the odds ratio are inconsistent. Please change the direction of the odds ratio (or change the sentence to "Girls, received less inhalation..."). DISCUSSION SECTION: 8. Please make a subsection for limitations. It is reader-friendly. CONCLUSION SECTION: 9. The conclusion section looks a part of discussion section. The author's conclusion is as follows?: "In children, the role of sex and gender on health is largely unknown and research assessing sex - specific differences is scare. Future studies should focus on the role of sex and gender in specific conditions and determine the influence of biological, as well as social and cultural factors." If so, what is the importance of this study? There are no summary of findings and implications.
--	---

VERSION 1 – AUTHOR RESPONSE

Reviewer(s)' Comments to Author:

Reviewer: 1

Reviewer Name: Rhonda J. Rosychuk

Institution and Country: University of Alberta, Canada

Please state any competing interests or state 'None declared': None

This study is interesting and important. It focuses on sex differences for children presenting at EDs, where there is a paucity of literature. The authors use data from multiple countries, which is a strength, and pool results through random effects meta-analysis. The large data set and analyses by general medical problems and trauma and injuries are also strengths. The manuscript is well written and the figures, tables, and appendices are appropriate.

We thank the reviewer for her review of our paper.

Specific comments:

1. There is a lack of clarity about the unit of the analysis. The authors sometimes reference children and sometimes reference ED visits. Is the unit of analysis the child or the ED visit? It would seem that the unit of analysis is the child, but I would expect that some of these children have more than one ED visit. The Methods section describes 119,209 consecutive ED visits and 1,771 children with incomplete data are removed. The Results section mentions 116,440 children: $16440+1771=118211 \neq 119209$ so there must be some children who have repeat ED visits. The ED visits from the same child would be correlated, and not be independent data. Hence the standard errors are likely a little too small because correlated data is unaccounted for. Given the large scale of the dataset, the resulting ORs and 95% CIs would not be altered very much. Perhaps authors can restrict to selecting one ED visit per patient so that they do not have to use statistical methods for correlated data.

We thank the reviewer for pointing out this important methodological issue and agree that the terminology we use is inconsistent. We would like to clarify that the unit of the analysis is the ED visits. In the manuscript, we have changed all the references from "children" to "visits".

Regarding the number of children in the analysis: beside the 1771 children with incomplete data we excluded children from Vienna with minor trauma ($n=681$ in the current analysis). We added this number to the Methods section and added a study flowchart to clarify the study's in- and exclusion criteria (Figure 1). The reason for exclusion is, that patients with a trauma or injury in Vienna are generally seen in the department of traumatology. Only a small subset of patients chooses to be referred to the "in-house paediatrician". Because this is only allowed for patients with MTS urgency Standard or Non-Urgent (categories 4 or 5), this represents a very specific subgroup. To avoid selection bias, we decided to exclude all trauma cases from the Vienna database. We have explained the rationale for this in the Methods section paragraph study settings and patient population: "*The General Hospital Vienna only included ED visits for medical complaints because the majority of trauma patients were seen in the department of traumatology. A small proportion of low urgent trauma cases were still seen in the ED by the "in-house paediatrician" and we excluded these remaining patients to reduce selection bias*". We would like to clarify that for reasons of consistency, we have used a different definition of "Trauma" which slightly changes the number of included patients ($n=268$, 0.2%) which has not affected our results.

Regarding the notion that some children have multiple ED visits, we agree with the reviewer that visits from the same child are not independent. Ideally, we would have adjusted for a patient-level variable in the analysis. Unfortunately, a variable indicating the patient identifier is not readily available in all of the ED settings. As the reviewer mentioned, considering the repeat visits as independent does not affect the size and direction of the effect estimates, but only the variances and therefore the width of

the confidence intervals.¹ Omitting the repeat visits from our database, however, could also introduce bias and would decrease the power of the study. Consequently, we have opted to keep the repeat visits in the study population but we acknowledge the potential limitation in this approach. We added a statement in the discussion section of the manuscript: *“Additionally, our cohort of ED visits included repeat visits from the same child which are not independent. In general, ignoring correlated data does not influence effect estimates such as odds ratios, but it can lead bias in the standard errors. By not taking into account the correlation of visits from the same subject we are likely to underestimate the standard errors of our effect sizes, potentially leading to wider confidence intervals.”* Additionally, we have mentioned this issue as a separate bullet point in the Strengths and limitations section: *“Our cohort of ED visits included repeat visits from the same child, which could have led to underestimation of the standard errors and potentially to wider confidence intervals.”*

2. Laboratory tests, imaging, medications, and admissions would be related to the presenting condition and not all of these diagnostic/management variables would be relevant for particular presenting conditions. It would seem to me that models for these outcomes should be adjusted by presenting condition as well. Would not the authors want to know if differences in diagnostics/management occur when adjusted for age, triage, and presenting condition? The subgroup analysis address this a bit but not completely.

We consider presenting condition as an important confounding variable and we agree with the reviewer that it would be better to adjust for this variable in the analysis. In line with a classification used in a previously published study², we categorized MTS' presentational flowcharts in ten groups representing the main paediatric clinical presentations in the ED. This variable was added, beside age and triage urgency, to the regression models assessing the relation between sex and diagnostics or management. We stated this in the methods section: *“Analyses were adjusted for age, triage urgency and clinical presentation, and boys were determined as the reference group”*. Moreover, we indicated the adjustment variables below the results in Table 3.

3. Analyses were adjusted for age and triage urgency. Was triage urgency considered as an outcome as well to see if there are sex differences in the assigning of triage urgency? This additional analysis could still be adjusted by age. It is unclear how subjective the triage urgency could be.

We considered triage urgency as an important confounder which could partly explain the differences in outcome between boys and girls. Therefore, we adjusted for it in the analysis. We agree with the reviewer that it would be interesting to have more information about the differences in triage urgency between boys and girls. Because we would like to avoid analysing triage urgency both as a confounder and an outcome, we have added descriptive analyses and additional figures that study the relation between gender and age and gender and clinical presentation, stratified by triage urgency. We added a separate paragraph in the results section describing these results: *“We further explored the relation between sex and triage urgency. The higher rate of boys' ED visits in childhood and girls' ED visits in adolescence was consistent among the different urgency categories. Also, the sex-specific differences in the different types of clinical presentation were similar in the high, intermediate and low urgency categories. An exception was the subgroup of uro- and gynaecological presentations where there were almost only boys prioritized as high urgency. Also notable was the subgroup of respiratory conditions that represents the largest population of high urgency visits with a 1.5 times higher proportion of boys compared to girls.”* The figures were added as appendix 4

4. Would be helpful to put the study period in the abstract or at least some indication of the general time period of data because the data from different sites may have been collected at different time periods.

We agree with the reviewer that this information is missing in the abstract and added the time period for data collection and range of study period per setting under the “Participants” heading: *“All consecutive paediatric ED visits of children under the age of 16 during the study period (8 to 36 months between 2012 and 2015)”*

5. I would have expected a bit more summary and/or comparison with the findings of other relevant studies rather than providing an appendix of search terms/citations.

We thank the reviewer for addressing this issue which was also raised by the editor and reviewer 3. In the introduction section, we have added examples of gender differences which have been found in adults, including references: *“In the area of emergency medicine, sex and gender-specific differences have been found in adults, in topics as broad as the clinical presentation and outcomes of acute myocardial infarction, the prevalence and survival of out-of-hospital cardiac arrest and the epidemiology of sports-related injuries”*.

Furthermore, we expanded the section on our literature review and discuss the remaining knowledge gaps. Particularly, we highlight that research on sex-specific differences in diagnosis and management is lacking: *“Of these, only two studies described differences between sexes in overall emergency department resource use or management. Both studies observed higher ED attendance rates in boys compared with girls, but did not address specific types of clinical presentations or evaluate differences in disease management or outcome. Other studies focused on specific disease groups such as trauma and injuries, mental health related conditions, and asthma. In these areas studies report conflicting results regarding differences in the rate of ED visits between boys and girls. Diagnosis and management were poorly studied. Only one study in children with asthma looked at differences in ED management and did not find a difference in treatment between boys and girls”*.

Finally, we added tables with details on the individual studies to the appendix (appendix 1).

6. It is not clear how the Manchester Triage System urgency category is defined. A reference is not provided and the flowchart in Appendix 3 does not indicate urgency.

We thank the reviewer for addressing this issue. In the methods section, paragraph determinants, we added a more detailed explanation about the MTS and how the urgency category is defined, including a reference to the original MTS book early in the paragraph. *“We used triage data to determine type of presenting problem and triage urgency. All participating hospitals used the Manchester Triage System (MTS). This flowchart-based emergency medical triage system is the most commonly used triage system in Europe. In the MTS, the triage nurse is required to select a flowchart for each patient, representing the chief complaint, such as Shortness of Breath or Wounds. Each flowchart consists of signs and symptoms named discriminators that are ranked by priority. The nurse then gathers information on the discriminators from top to bottom. Selection of a discriminator allocates the patient to the related urgency category ranging from “Immediate” (0 minutes maximum waiting time) to “Non-urgent” (240 minutes maximum waiting time).”*

7. The grouping of the disposition categories are unclear and/or unjustified. ICU and mortality at ED are combined and I would think that these two outcomes are different. Did the authors mean that all ICU patients died in the ED?

Because mortality at the ED in our study population is very low (n=16, 0.01%), we choose to combine mortality and ICU admission both as markers of high urgency. We agree with the reviewer that both are different categories: mortality at the ED indicates that a patient died while being in the ED, before transfer to another department was possible. In table 1, which we added to the main manuscript, we have now provided numbers for both categories separately.

I do not see a CONSORT checklist submitted in the documents I can see online.

We would like to confirm to the reviewer that we added a CONSORT checklist to our submission. We have added the checklist as an appendix to our decision letter so it will be visible for the reviewer.

Reviewer: 2

Reviewer Name: Taneisha Wilson

Institution and Country: Brown University, USA

Please state any competing interests or state 'None declared': None declared

We thank the reviewer for her review of our paper

Page 1: Line 21/27 we use girls and boys here interchangeably with male/female for line 21 would suggest maybe use boy/girl as parenthetical form

We thank the reviewer for addressing this discrepancy. We changed all instances where we used male/female in the manuscript as well as the tables into boys/girls

Page 9: I would suggest a flow diagram here. Line 51, again would use boy vs. girl instead of male-female based on your demographic population.

We thank the reviewer for this suggestion and added a flow diagram of our study population as Figure 1. Also the words male and female were changed into boys and girls.

Page 12: line 29-30, thought boys were the reference variable and girls the comparator. If this is the case I would reframe this (Appendix 6).

We thank the reviewer for pointing out this error. We rewrote the results section and ensured that we now consistently describe the odds for girls as compared with boys. The statement regarding the inhalation medication is now as follows: "*Remarkably, there was a large difference in the use of inhalation medication in the subgroup of children with respiratory conditions. In this subgroup, girls received less inhalation medication (pooled OR 0.76, 95% CI 0.70-0.83)*".

Also, assuming that the age subgroups include >12=adolescent based on page 12 line 53, but this is not explicitly stated.

We agree with the reviewer that it is not clearly stated that our study includes children as well as adolescents. We added to our methods section paragraph "Study settings and patient population" a statement where we explicitly mention that our study includes adolescents as well: "*In the TRIAGE study, 119,209 consecutive ED visits of children and adolescents under the age of 16 years were included*". Furthermore, we have attempted to avoid using the word "children" and used ED visits instead.

Reviewer: 3

Reviewer Name: Tadahiro Goto

Institution and Country: The University of Tokyo, Japan

Please state any competing interests or state 'None declared': None declared

I appreciate the opportunity to review this manuscript. In this large, multicenter and international cohort study of 116,440 emergency department visits made by children reports that sex is associated with differences in disease presentation and management. While this study might provide some information for clinicians and researchers, several concerns should be addressed.

We thank the reviewer for his critical review of our manuscript.

General comments

=====

1. While this paper is well-written and the data might be informative, I am not sure the importance of the study findings and its implications. What is the knowledge gap? What will be changed by the study findings? The descriptive findings are interesting and should be the important basis for the future investigation. However, the importance and clinical implications of the associations (e.g., boys received more inhalation medication) are not clear.

We thank the reviewer for addressing this crucial issue and we agree that the importance of the study findings need more emphasis in our manuscript. We have rewritten the introduction and discussion sections of the paper in light of the reviewer's comments.

First, we would like to state that there is a lack of studies on gender differences in children, and in paediatric emergency medicine in particular. In adult medicine, studies have started to explore sex-specific differences with surprising results. These findings of differences between man and women have changed healthcare practice and improved care for women. We strongly believe that it is important to study whether there are sex-specific differences in childhood as well. Given that the few studies that have been conducted on this subject are conducted in different areas with often conflicting results, we propose that further and more rigorous research is needed. Our study is the first to address this topic on such a large scale and provide information on the broad population of children visiting the emergency department. We have extended our introduction section to make this argument clearer and we will also address this more specifically in the next reviewer comment.

Second, we acknowledge that our analyses are exploratory and that the associations need further study. However, a starting point for research on sex-specific differences in paediatric emergency medicine is needed. Our study invites researchers to assess differences in diagnostics in boys and girls. For example, do girls and boys with the same condition have a different way of presenting? Can we find evidence that provider attitude differs when managing a boy or a girl in the ED? Are there clues for a pathophysiologic difference between boys and girls presenting with shortness of breath or fever?

We have rewritten our discussion section and focus on the two main findings. The first is the higher rate of ED visits from boys and the higher use of inhalation medication. This has been previously described and we discuss the available literature. More importantly, we found evidence that more diagnostic tests were performed in girls. This, to the best of our knowledge, has not been described before. We already mention some possible explanations, and we extended this paragraph with some areas for future research to make the implications of our research more visible: *"Our analyses are exploratory and our findings require further research. Future studies are needed that address sex-specific differences in the conducting of diagnostic tests. These studies should aim to elucidate whether differences in the need for diagnostics are due to pathophysiological mechanisms and differences in disease course, whether girls present signs and symptoms differently compared with boys, or whether there are other sociocultural factors responsible."*

Specific comments

INTRODUCTION:

2. Please clarify the knowledge gap. While there are 22-relevant articles in this area, the knowledge gap remains unclear.

We thank the reviewer for addressing this issue which was also raised by the editor and reviewer one. We would like to argue that 22 articles is very little in a topic as broad as emergency medicine and we agree with the reviewer that this requires further clarification and explanation. In the introduction, we expanded the section on our literature review and discuss the remaining knowledge gaps.

Particularly, we highlight that research on sex-specific differences in diagnosis and management is lacking: *"Of these, only two studies described differences between sexes in overall emergency department resource use or management. Both studies observed higher ED attendance rates in boys compared with girls, but did not address specific types of clinical presentations or evaluate differences in disease management or outcome. Other studies focused on specific disease groups such as trauma and injuries, mental health related conditions, and asthma. In these areas studies report conflicting results regarding differences in the rate of ED visits between boys and girls. Diagnosis and*

management were poorly studied. Only one study in children with asthma looked at differences in ED management and did not find a difference in treatment between boys and girls". As a reference, we added tables with details on the individual studies to the appendix (appendix 1)

Furthermore, we reworded the aim of our study in the final paragraph of the introduction: "The aim of the current study is to assess the role of sex in the clinical presentation, diagnostics and management in the general population of children attending the emergency department (ED). Through this study, we aim to gain more insight in sex-specific differences in paediatric emergency medicine and identify areas for future research".

METHODS SECTION:

3. While authors excluded patients with missingness, the lack of clinical information might be not at random (e.g., critically-ill children). Are there any potential bias due to the missingness?

In total we excluded 2% of patients and we agree with the reviewer that this requires further detail. First, we added an additional flow diagram to the manuscript to provide more detail regarding the in- and exclusion of patients (new Figure 1). Moreover, we analysed differences between the sexes in the ED visits that were excluded, but found that this difference was not significant. We added this analysis to the results section: "There was no significant difference between the sexes in the proportion of excluded ED visits (Pearson's Chi-Square(1) = 0.08, p-value 0.77." Since the proportion of excluded patients in our study is very small (rule of thumb <5%) and is not differential between the sexes we do not believe it could have had a great effect on our results. Therefore, we believe it is acceptable to exclude the patients with missing data instead of using more sophisticated methods such as multiple imputation.

4. Why authors used laboratory tests and imaging as an outcome?

We thank the reviewer for raising this issue. In our study we aimed to evaluate both diagnostics and management. Diagnostic tests are interesting because they reflect the need for further diagnostic certainty and are important resources in the ED. We selected laboratory tests and imaging because they are the two main categories of diagnostics. We rephrased the paragraph discussing our outcome measures in the methods section: "We selected laboratory tests and imaging as important markers for diagnostics, and inhalation medication, intravenous medication or fluids and hospital admission as markers for disease management."

5. Why authors choose fever and shortness of breath for the subgroup analysis? I understand that these two complaints are common, but it looks arbitral when there are no supporting literature or rational.

We thank the reviewer for sharing his thoughts on the selection of subgroups. We agree that this was an intuitive decision and we have now chosen to select subgroups based on the clinical categories that we created. These clinical categories, as explained in the answer to the next reviewer question, are based on the triage system's presentational flowcharts and in line with a previous publication. We did not consider the largest subgroup of children, those presenting with general malaise, because this is a very broad and aspecific population of children. Consequently, we selected the three largest subgroups (trauma or musculoskeletal, diarrhoea and vomiting and respiratory symptoms). Furthermore, we added the subgroup of children with fever. This subgroup is not a separate group in the ten clinical categories because there is overlap with many of the different presentations (dermatological, respiratory, gastrointestinal, general malaise, limb problems etc.). However, we consider children with fever a very important subset of children in the ED with challenging diagnostic and therapeutic decisions. We have added a motivation for the selection of the different subgroups: "We performed subgroup analyses in subgroups of children presenting with trauma or musculoskeletal problems, children presenting with fever, children presenting with gastro-intestinal

problems and children presenting with shortness of breath. (...). We selected these four clinical presentations because they were among the largest subgroups of patients and represent important clinical entities in children."

In addition, because our rationale in the choice of subgroups, we added the subgroup of trauma patients as a separate subgroup, unlike in the previous version of the manuscript where we made the distinction from the beginning.

6. Authors performed subgroup analysis based on patient's chief complaints. This implies that the association between sex and outcomes may differ across chief complaints. If so, why authors did not include chief complains as an adjustment variable?

We consider presenting condition as an important confounding variable and we agree with the reviewer that it would be better to adjust for this variable in the analysis. In line with a classification used in a previously published study², we categorized MTS' presentational flowcharts in ten groups representing the main paediatric clinical presentations in the ED. This variable was added, beside age and triage urgency, to the regression models assessing the relation between sex and diagnostics or management. We stated this in the methods section: "*Analyses were adjusted for age, triage urgency and clinical presentation, and boys were determined as the reference group*". Moreover, we indicated the adjustment variables below the results in Table 3.

RESULTS SECTION:

7. Page 12, line 29-31. "Boys, on the other hand, received more inhalation medication (pooled OR 0.75...)." The subject and the odds ratio are inconsistent. Please change the direction of the odds ratio (or change the sentence to "Girls, received less inhalation...").

We thank the reviewer for pointing out this error. We rewrote the results section and ensured that we now consistently describe the odds for girls as compared with boys. The statement regarding the inhalation medication is now as follows: "*Remarkably, there was a large difference in the use of inhalation medication in the subgroup of children with respiratory conditions. In this subgroup, girls received less inhalation medication (pooled OR 0.76, 95% CI 0.70-0.83)*".

DISCUSSION SECTION:

8. Please make a subsection for limitations. It is reader-friendly.

We thank the reviewer for addressing this issue. We added to the discussion a separate section on limitations and a separate conclusion (see reviewer comment 9) to improve readability.

CONCLUSION SECTION:

9. The conclusion section looks a part of discussion section. The author's conclusion is as follows?: "In children, the role of sex and gender on health is largely unknown and research assessing sex -specific differences is scare. Future studies should focus on the role of sex and gender in specific conditions and determine the influence of biological, as well as social and cultural factors." If so, what is the importance of this study? There are no summary of findings and implications...

We agree with the reviewer that there is no separate conclusion, as we finish our discussion section with future perspectives. We have added a separate concluding paragraph including a summary of the findings and short statement on the need for future studies: "*Our study found that in childhood boys more often present to the ED compared with girls, while in adolescence this ratio is reversed. The higher need for inhalation medication in boys may represent a higher susceptibility for or a more severe course of respiratory infections. Unexpectedly, girls receive more diagnostic tests compared with boys. Future studies should focus on the role of sex and gender in specific conditions and determine whether there are pathophysiological differences in disease course and severity, whether*

girls present signs and symptoms differently or whether there are social and cultural factors responsible.”

VERSION 2 – REVIEW

REVIEWER	Rhonda Rosychuk University of Alberta
REVIEW RETURNED	24-Mar-2020

GENERAL COMMENTS	Sex-Specific Differences in Children Attending the Emergency Department: Prospective Observational Study Zachariasse et al Thank you to the authors for providing a comprehensive revision that addresses my comments well. I still have a couple of opportunities for clarification.  1. The authors have clarified that the unit of the analysis is the ED visit. I recognize that without patient-specific IDs the correlated data cannot be properly adjusted for in the calculation of standard errors.  a. Could the authors know any aggregate data from these centres about the distribution of repeated ED visits? For example, historical data might suggest that less than 5% of patients have more than 1 ED visit in the same year. That would be helpful for the description and perhaps provide a better sense of how biased the SEs might be. b. Please put in the data analysis section a statement to the effect that ED visits were treated as independent in analyses although some children likely had repeated visits and those visits could not be distinguished in this study (could be put before the pooling statement). c. I appreciate the limitation saying “Our cohort of ED visits included repeat visits from the same child, which could have led to underestimation of the standard errors and potentially to wider confidence intervals”. I think this statement may not be clear for all readers. It could be interpreted as saying that the presented CIs are wider than they really should be. The issue is that the presented CIs could be smaller than they should be, leading to potentially falsely concluding statistical significance. So maybe something like “Our cohort of ED visits included repeat visits from the same child, which could have led to underestimation of the standard errors. Such underestimation could potentially lead to smaller confidence intervals than would be obtained when analyses are adjusted for repeat visits from the same child.” 2. I see that the authors have now changed from the terminology of male/female to boy/girl. I have come to understand sex as the biological attributes (usually referred to as male/female) and gender as the socially constructed roles, behaviours, expressions and identities
---

	(often referred to as boy/man, girl/woman, and other categorizations). See for example: https://cihr-irsc.gc.ca/e/48642.html . I would think that the data sources records sex and not gender, so references to male/female may be more appropriate than boy/girl. There may be disagreement on the terminology to use across the world and the Editor may wish to weigh in on the best approach for the journal.
--	--

REVIEWER	Tadahiro Goto The University of Tokyo, Japan
REVIEW RETURNED	19-Mar-2020

GENERAL COMMENTS	My concerns have been appropriately addressed. I would applaud the authors' excellent work. Thanks.
---

VERSION 2 – AUTHOR RESPONSE

Reviewer 2's comments:

Page 1: Line 21/27 we use girls and boys here interchangeably with male/female for line 21 would suggest maybe use boy/girl as parenthetical form

Page 9: I would suggest a flow diagram here. Line 51, again would use boy vs. girl instead of male-female based on your demographic population.

We thank the editor for commenting on this issue. We agree with reviewer 1 that our variable refers to the patients' sex and thus the terms male/female is more appropriate. On the other hand, we agree with reviewer 2 that based on the age of our population the terms boy/girl better describes the population.

Therefore, we changed the instances where we refer to the general notion of sex into male/female (for example, abstract heading "main outcome measure" and methods paragraph "determinants"). The instances where we refer to our specific population, we continue to use the term boy/girl, as we did in our initial submission.

If the editor prefers to use the term male/female throughout the manuscript, even in the instances where we did not initially use these terms, we will of course change it.

Reviewer(s)' Comments to Author:

Reviewer: 3

Reviewer Name: Tadahiro Goto

Institution and Country: The University of Tokyo, Japan Please state any competing interests or state 'None declared': None declared

Please leave your comments for the authors below My concerns have been appropriately addressed. I would applaud the authors' excellent work. Thanks.

We thank the reviewer for his review of our revised manuscript and for his comments which have helped to improve the paper.

Reviewer: 1

Reviewer Name: Rhonda Rosychuk

Institution and Country: University of Alberta Please state any competing interests or state 'None declared': None declared.

Thank you to the authors for providing a comprehensive revision that addresses my comments well.

We thank the reviewer for reviewing the revised version of our paper.

I still have a couple of opportunities for clarification.

1. The authors have clarified that the unit of the analysis is the ED visit. I recognize that without patient-specific IDs the correlated data cannot be properly adjusted for in the calculation of standard errors.

a. Could the authors know any aggregate data from these centres about the distribution of repeated ED visits? For example, historical data might suggest that less than 5% of patients have more than 1 ED visit in the same year. That would be helpful for the description and perhaps provide a better sense of how biased the SEs might be.

We have explored data regarding revisits from a single hospital (Maasstad Hospital), because in this database we had a coded patient identifier readily available. The data consisted of 10,482 ED visits by 8253 children. 6695 children (81%) had 1 ED visit during the study period, 1411 (17%) had two or three ED visits and 147 (2%) had more than three ED visits. To allow each patient to occur in the dataset only once, we split the dataset into primary visits (n=8253, 79%) and repeat visits (n=2229, 21%). The proportion of female visits in the primary visit subset was 44%, and in the repeat visit subset 43%. This difference was not statistically significant according to Pearson's Chi-square test ($\chi^2 = 0.62$, $p = 0.43$). Thus, we did not observe differences between the sexes in the rate of revisits.

The proportion of revisits is in line with previous findings from a study, based on national sample of data from the USA's Medical Expenditure Panel Survey, 2010 to 2014 examining annual ED utilization of children age 0 to 17 years. This study found that among children with ED visits, 21% had two or more visits to the ED in a 1-year period.¹

We are not aware of any method to estimate to quantify the bias in standard errors. Therefore we only added data on the possible proportion of revisits to the limitations paragraph of the Discussion: "A previous study shows a revisit rate of approximately 21% in a representative sample of data from U.S. EDs".

b. Please put in the data analysis section a statement to the effect that ED visits were treated as independent in analyses although some children likely had repeated visits and those visits could not be distinguished in this study (could be put before the pooling statement).

We thank the reviewer for the suggestion to add some clarification in the methods section and we added the following statement to the data analysis paragraph in the Methods section: "*ED visits were treated as independent in the analyses although some children likely had repeated visits and those visits could not be distinguished in this study*".

c. I appreciate the limitation saying "Our cohort of ED visits included repeat visits from the same child, which could have led to underestimation of the standard errors and potentially to wider confidence intervals". I think this statement may not be clear for all readers. It could be interpreted as saying that the presented CIs are wider than they really should be. The issue is that the presented CIs could be smaller than they should be, leading to potentially falsely concluding statistical significance. So maybe something like "Our cohort of ED visits included repeat visits from the same child, which could have led to underestimation of the standard errors. Such underestimation could potentially lead to smaller confidence intervals than would be obtained when analyses are adjusted for repeat visits from the same child."

We agree with the reviewer that this statement was not clearly phrased and we changed the sentence in the limitations paragraph according to the suggestion: "*By not taking into account the correlation of visits from the same subject we are likely to underestimate the standard errors of our effect sizes. Such underestimation could potentially lead to smaller confidence intervals than would be obtained when analyses are adjusted for repeat visits from the same child*".

2. I see that the authors have now changed from the terminology of male/female to boy/girl. I have come to understand sex as the biological attributes (usually referred to as male/female) and gender as the socially constructed roles, behaviours, expressions and identities (often referred to as boy/man, girl/woman, and other categorizations). See for example:

[https://cihrirsc.](https://cihrirsc.gc.ca/e/48642.html)

[gc.ca/e/48642.html](https://cihrirsc.gc.ca/e/48642.html). I would think that the data sources records sex and not gender, so references to male/female may be more appropriate than boy/girl. There may be disagreement

on the terminology to use across the world and the Editor may wish to weigh in on the best approach for the journal.

We agree with the reviewer that our variable refers to the patients' sex and thus the terms male/female is more appropriate. We changed the wording in the manuscript based on other reviewer's comments that the term boy/girl better describes our population.

Therefore, we have changed the instances where we refer to the general notion of sex into male/female (for example, abstract heading "main outcome measure" and methods paragraph "determinants"). The instances where we refer to our specific population, we continue to use the term boy/girl, as we did in our initial submission.

References

1. Schlichting LE, Rogers ML, Gjelsvik A, et al. Pediatric Emergency Department Utilization and Reliance by Insurance Coverage in the United States. Acad Emerg Med 2017;24(12):1483-90.

VERSION 3 – REVIEW

REVIEWER	Rhonda Rosychuk University of Alberta
REVIEW RETURNED	06-Jul-2020
GENERAL COMMENTS	I am satisfied with the changes and responses to reviewer comments.